



# Single-photon laser-induced fluorescence detection of nitric oxide at sub-parts per trillion mixing ratios

Andrew W. Rollins[1], Pamela S. Rickly[1,2], Ru-Shan Gao[1], Thomas B. Ryerson[1], Steven S. Brown[1], Jeff Peischl[1,2], and Ilann Bourgeois[1,2]

[1]NOAA Earth System Research Laboratory, Chemical Sciences Division
[2]Cooperative Institute for Research in Environmental Sciences, University of Colorado Boulder

**Correspondence:** Andrew Rollins (andrew.rollins@noaa.gov)

**Abstract.** We describe a newly developed single-photon laser-induced fluorescence sensor for measurements of nitric oxide (NO) in the atmosphere. Rapid tuning of a narrow-band laser on and off of a rotationally resolved NO spectral feature and detection of the red-shifted fluorescence provides for interference-free direct measurements of NO with a detection limit of 1 pptv for 1 second of integration, or 0.3 pptv for 10 seconds of integration. The instrument was deployed on the NASA DC-8 aircraft during the NASA FIREX-AQ experiment during July - September of 2019 and provided more than 140 hours of NO measurements over 22 flights, demonstrating the ability of this instrument to operate routinely and autonomously. Comparisons with a seasoned chemiluminescence sensor during FIREX-AQ in a variety of chemical environments provides validation and confidence of the accuracy of this technique.

## 1 Introduction

Nitric oxide (NO) is central to radical chemistry in Earth's atmosphere. In the troposphere the catalytic reaction of NO with the hydroperoxy and organic peroxy radicals

$$NO + RO_2/HO_2 \rightarrow NO_2 + RO/HO \tag{1}$$

is frequently the rate-limiting step for the production of tropospheric ozone ($O_3$), and is the reason why anthropogenic emissions of NO result in a buildup of $O_3$ pollution. Oxidation of NO ultimately also results in the formation of nitric acid, and consequently nitrate aerosols and nitrogen deposition. NO has an important control over the partitioning of atmospheric $HO_x$ ($HO_x$ = OH + $HO_2$) due to the reaction between NO and the hydroperoxyl radical (Gao et al., 2014). Stratospheric $NO_x$ ($NO_x$ = NO + $NO_2$) is important for suppressing concentrations of chlorine monoxide (ClO) which leads to rapid destruction of $O_3$ (Fahey et al., 1993; Solomon, 1999). Increases in stratospheric aerosols lead to more rapid heterogeneous conversion of $NO_x$ into nitric acid and consequently an increase in ClO; a potential by-product of future solar radiation management efforts (Tilmes et al., 2018).

Active fields of atmospheric research seek to understand radical chemistry cycling in low NO regimes. Hydroxyl radical budgets in forested environments where NO measurements are near 10 parts per trillion (pptv, $10^{-12}$ mol mol$^{-1}$) remain incompletely understood, with measurements of OH frequently exceeding calculated OH based on known chemistry (Rohrer



et al., 2014). Autooxidation of organic compounds in low-NO environments is increasingly recognized as a key source of
highly-oxidized/low-volatility organic compounds in the atmosphere (Crounse et al., 2013). Measurements of the ratio of NO
to nitrogen dioxide ($NO_2$) in the upper troposphere (UT) frequently cannot be reconciled with models (Cohen et al., 2000;
Silvern et al., 2018). Zhao et al. (2019) show that differences in background NO at the single pptv level are responsible for
differences in global modeled OH on the order of 50%. The ability to measure atmospheric NO at very low mixing ratios and
with low uncertainty will be crucial to address these and other questions in atmospheric chemistry research for the foreseeable
future.

Almost all of the research in the past two decades associated with direct detection of atmospheric NO has relied on the
chemiluminescence (CL) detection technique (Ridley and Howlett, 1974). In this method, a sample of air is mixed with a
high concentration ($\sim 1\%$) of $O_3$ resulting in the formation of electronically excited nitrogen dioxide ($NO_2^*$) which produces
detectable luminescence in the near-infrared. This technique while being quite precise ($\sim$ 5-10 pptv detection limit), has
significant drawbacks. These include: potential positive or negative interferences from other species including a variety of
organic compounds (Drummond et al., 1985), precision limits on the order of 5-10 pptv, significant instrumental background
levels (10-100 pptv equivalent) which might reduce accuracy, reliance on consumables including pure oxygen and dry ice,
production of %-levels of ozone which is toxic and must be exhausted from the instrument.

An alternative technique which has been explored previously is laser-induced fluorescence. Due to the NO absorption cross
sections in the deep ultraviolet region of greater than $10^{-16}$ cm$^2$ molecule$^{-1}$ and moderately high fluorescence quantum yield
($\sim$ 10%), it should be expected that very precise NO measurements could be made using such a technique. Atmospheric
measurements using both single-photon and two-photon excitation schemes were demonstrated by Bradshaw et al. (Bradshaw
et al., 1982, 1985) and Bloss et al. (Bloss et al., 2003). The single-photon excitation scheme employed by Bradshaw et al. used
a dye-based laser system near 226 nm to excite the $v' = 0$ manifold of the $A^2\Sigma$ electronic state and observe the red-shifted
fluorescence emission from relaxation near 259 nm ($v' \rightarrow v'' = 0 \rightarrow 3$). This system had a reported detection limit of 28 pptv
for 1 minute of integration. The two-photon scheme (Bradshaw et al., 1985) involved further excitation of NO that had been
pumped into the $A^2\Sigma$ state using a second photon at 1.06 $\mu$m to promote NO into the $D^2\Sigma$ state. The highly excited NO would
emit blue shifted fluorescence at 187 nm, which had the advantage of being detected on a near-zero background. The reported
detection limits for this technique were 1 pptv given 5 minutes of integration, or 10 pptv for 30 seconds of integration (Bradshaw
et al., 1985). To our knowledge, the two-photon scheme developed by Bradshaw et al. is the only NO-LIF measurement that
has been utilized extensively for atmospheric measurements including on aircraft, and this technique was last successfully used
during the NASA TRACE-P experiment in 2001. Field comparisons of the two-photon excitation scheme with multiple CL
instruments demonstrated similar performance for the two techniques (Hoell et al., 1987). Bloss et al. (2003) used a frequency
quadrupled Ti:Sapphire laser system to excite NO near 226 nm and detected broadband red-shifted fluorescence at 240-390
nm. Although Bloss et al. did not state a detection limit for their prototype instrument, they calculated that a detection limit of
0.07 pptv for 60 s of signal integration might be achievable with improvements in their system. More recently Mitscherling et
al. (Mitscherling et al., 2007, 2009) reported investigation of the use of single photon LIF detection of NO for human breath
analysis by pumping the $A^2\Sigma \leftarrow X^2\Pi$ transition both near 226 nm ($v' = 0$) and near 215 nm ($v' = 1$).





In this work we report on the recent development of a new single photon LIF sensor that pumps the $A^2\Sigma(v'=1) \leftarrow X^2\Pi(v''=0)$ vibronic transition near 215 nm and observes the resulting red shifted fluorescence from $\sim$ 255 - 267 nm. The present system is distinguished from previous efforts to use LIF to measure atmospheric NO primarily because we use a fiber-amplified laser system. This system has numerous advantages, including: 1) laser linewidth that is sufficiently narrow to resolve the Doppler broadened NO spectrum at room temperature and thereby achieve high signal levels and distinguish the NO isotopologues, 2) laser repitition rate high enough to enable single-photon counting of the fluorescence signal, 3) size, weight and environmental robustness allowing for practical and routine integration onto airborne research platforms. The current version of this system has a detection limit ($2\sigma$) of $\sim$ 1 pptv for 1 second of integration or $\sim$ 0.3 pptv for 10 seconds of integration. Uncertainty in the instrument zero is demonstrated to be less than 0.2 pptv. The instrument was integrated onto the NASA DC-8 aircraft during the NASA/NOAA FIREX-AQ experiment (Fire Influence on Regional to Global Environments Experiment - Air Quality) during 2019. Here we describe the instrument and its performance during this initial deployment.

## 2 NO-LIF Detection

### 2.1 Instrument Description

In this section we describe the physical components of the LIF instrument. Subsequent sections discuss details of the NO spectroscopy and instrument performance.

The laser and optical detection system used here are based on that originally described by Rollins et al. (2016) for measurements of sulfur dioxide. Subsequent to that work, important changes have been made to the design of the fiber laser system, and therefore a complete description is provided here.

The laser wavelength is controlled by modulating the current of a distributed feedback laser (DFB). This fiber-coupled DFB laser can provide up to 50 mW CW optical power in a single-mode polarization-maintaining fiber and can be current tuned in the range of 1074 - 1076 nm with a nominal linewidth of 10 MHz. The DFB output is chopped to make pulses of 2-3 ns in duration with a 320 kHz repetition rate using a fiber-coupled electro-optic modulator with 10 GHz of switching bandwidth and an extinction ratio of 40 dB. The $\sim$ 40 pJ pulses are amplified in a multi-stage ytterbium-doped fiber amplifier to near 5 $\mu$J and then exit the fiber-amplifier system through an end-capped fiber and the beam is collimated to a diameter of 350 $\mu$m.

The pulses pass through three nonlinear crystals to produce the fifth harmonic near 215 nm with a yield of $\sim$ 1%. The first crystal is a type-II phase matched KTP crystal (3 x 3 x 10 mm) producing the second harmonic at 537.5 nm. The second crystal is a type-I phase matched LBO crystal (3 x 3 x 10 mm) which mixes the 537.5 nm light with the residual 1075 nm light to produce the third harmonic at 358.3 nm. A custom dual-wavelength waveplate ($\lambda/2$ @ 537.5 nm, $\lambda$ @ 358.3 nm) rotates the residual second harmonic beam to be parallel with the third harmonic beam. A 40 mm focal length lens is then used to slightly focus the beams into a type-I phase matched BBO crystal (3 x 5 x 20 mm) producing the fifth harmonic near 215 nm. A Pellin-Broca prism separates the harmonics and all of the light is trapped except for the 215 nm beam, which is steered into the LIF cell.





Figure 1 depicts the layout of the free-space portion of the optical system and the NO detection system. The 215 nm beam, which is typically near 1 mW, passes through the fluorescence sample cell a single time. The beam is then split about 10/90 with 10% of the power entering a solar-blind power monitoring phototube (Hamamatsu R6800U-01) and 90% passing through a reference fluorescence cell and into a second phototube. The reference cell typically has a constant flow of 500 ppbv NO

flowing at 50 sccm and the exhaust of the cell is tied to the exhaust of the sample cell such that both cells are at pressures within 0.5 hPa during measurements. Inside each cell, a fused silica lens with numerical aperture of 0.5 collects fluorescence light from the center of the cell, which then passes through a $260 \pm 8$ nm bandpass filter (Semrock FF01-260/16) and is then imaged onto the photocathode of a photomultiplier tube (PMT) module operated in single-photon counting mode (Hamamatsu H12386-113). In the reference cell, it is necessary to reduce the signal on the PMT to maintain a response that is linear with

changes in laser power. Therefore a neutral density filter with a typical optical depth of 1.0 is placed between the bandpass filter and the PMT.

The sample flow and pressure are controlled by two custom stepper-motor controlled butterfly valves that are designed to minimize pressure drop through the system and potential sampling artifacts (Gao et al., 1999). The inlet valve is machined out of PEEK material (polyether ether ketone) and the exhaust valve out of stainless steel. The inlet valve is servo controlled

to a mass flow meter that measures the exhaust of the sample cell. The exhaust valve which is located after the point where the sample and reference cell exhausts are tied together, is servo controlled to a pressure transducer measuring the pressure immediately downstream of the sample cell. During flight operation the pressure and flow are typically maintained to within a 1% range of their setpoints over the entire altitude range encountered. Flow through the reference cell is controlled using a pair of mass flow controllers to mix zero-air and gas from a NO standard gas cylinder.

Data collection and instrument control are performed using a National Instruments compact RIO data system. This system incorporates a field-programmable gate array which is used to control the timing of the laser and photon detection gating with a precision of 5 ns. The instrument that was designed for operation on the NASA DC-8 aircraft houses the detection and reference cells, gas deck and data system in one enclosure (55 x 43 x 21 cm) and the fiber laser components in a second enclosure (43 x 43 x 5 cm). In total, these occupy 26 cm of vertical rack space in a standard instrumentation rack, and weigh 31 kg. Typically,

a vacuum scroll pump (Agilent IDP-3) and a small calibration gas bottle are installed in the rack adjacent to the instrument. These components take another 21 cm of vertical rack space and weigh 19 kg.

## 2.2 NO spectroscopy

Figure 2 illustrates the relevant NO electronic potential energy surfaces and the LIF scheme used in this work. We pump the $A^2\Sigma(v'=1) \leftarrow X^2\Pi(v''=0)$ transition near 215 nm, and observe the resulting red-shifted fluorescence from the $A^2\Sigma(v'=$

$1) \rightarrow X^2\Pi(v''=4,5)$ transitions.

The rovibronic spectrum of NO, especially in the 'gamma bands' ($A^2\Sigma \leftarrow X^2\Pi$), has been the subject of numerous previous studies (see Mitscherling (2009) and references therein). Figure 3 illustrates the absorption spectrum of NO. Line-by-line resolved spectra shown here have been calculated using the PGOPHER software package (Western, 2017). For the simulation of $^{14}N^{16}O$ absorption spectra we use the spectroscopic data reported by Danielak et al. (1997) and Murphy et al. (1993).





The tunable laser used here pumps NO near 215 nm from the ground $X^2\Pi(v''=0)$ state into the $A^2\Sigma(v'=1)$ state. Using this excitation has multiple advantages over the $A^2\Sigma(v'=0) \leftarrow X^2\Pi(v''=0)$ excitation scheme pumping at 226 nm. First is that the absorption cross section ($\sigma$) for NO is about twice as high in the $1 \leftarrow 0$ transition compared to $0 \leftarrow 0$. Second, the additional vibrational energy provides a more significant shift in the spectra of the various NO isotopologues, making them more easily distinguished spectroscopically. The origin of the $A^2\Sigma(v'=1) \leftarrow X^2\Pi(v''=0)$ transition for $^{14}N^{16}O$ is 46 cm$^{-1}$

and 70 cm$^{-1}$ higher in energy than those for the $^{15}N^{16}O$ and $^{14}N^{18}O$ isotopologues. Third, 215 nm can be produced using the fifth harmonic of a ytterbium-doped fiber amplifier system operating at 1075 nm, whereas 226 nm cannot currently be produced using such a system. In addition, excitation at 226 nm has the potential to produce spurious signal from fluorescence of SO$_2$, while 215 nm is a minimum in the SO$_2$ absorption cross section, and the SO$_2$ fluorescence quantum yield here is less than 3% (Hui and Rice, 1973).

Figure 4 shows the expected fluorescence emission spectrum based on the Franck-Condon factors (Scheingraber and Vidal, 1985; Danielak et al., 1997). Excluding the emission at $A^2\Sigma(v'=1) \rightarrow X^2\Pi(v''=0)$, which cannot be distinguished from Rayleigh scatter, fluorescence from the $A^2\Sigma(v'=1)$ state is expected to peak at $v''=4$ (255 nm) or $v''=5$ (267 nm) although it should be possible to collect significant signal from any of $v''=3 \sim 8$ (Scheingraber and Vidal, 1985). Multiple detection bandpass filters were tested to optimize the signal:noise for NO detection. The signals obtained with filters collecting the $1 \rightarrow 4$,

$1 \rightarrow 5$, and $1 \rightarrow 6$ transitions scaled relative to each other as expected with the Franck-Condon factors for those transitions. Of the filters tested, the one found to produce the lowest detection limit is a filter centered at 260 nm with a full width of 16 nm (see Fig. 4). This has 63% transmission at the $1 \rightarrow 4$ transition (255 nm) and 69% transmission at $1 \rightarrow 5$ (267 nm) while completely rejecting laser Rayleigh and Raman scatter from N$_2$ and O$_2$. While operating the detection cell near 80 hPa, typical background using this filter is 10 counts s$^{-1}$ with 1 mW laser power. Of this background, about 1 count s$^{-1}$ is a dark count from

the detector. Using a filter to additionally collect the $1 \rightarrow 6$ emission increased the signal by 4.5 counts s$^{-1}$ mW$^{-1}$ pptv$^{-1}$, but also increased the background to more than 350 counts s$^{-1}$ mW$^{-1}$ which would significantly degrade the detection limit. We expect that background levels would only further increase at longer collection wavelengths while collecting the fluorescence from $v''=3$ at 244 nm would likely increase the signal without substantial increases in the background.

### 2.3   LIF Signal

The anticipated LIF signal ($S$, counts s$^{-1}$) is proportional to the product of the NO excitation rate $E(\nu)$ (s$^{-1}$), the fluorescence quantum yield $\phi$, and the fluorescence collection efficiency of the detection system $\Omega$.

$$S = E(\nu) \cdot \phi \cdot \Omega \qquad (2)$$

In the optically thin regime, $E(\nu)$ can be approximated as the product of the convolution of the molecular absorption cross section ($\sigma(\nu)$ (cm$^2$ molecule$^{-1}$)) with the normalized laser spectral distribution ($\Lambda(\nu)$), the concentration of NO in the sample

cell ($n$, molecule cm$^{-3}$), the volume within the sample cell that is illuminated and imaged onto the PMT ($V$, cm$^3$) and the



laser photon flux in the sample volume ($\Phi$, photons s$^{-1}$ cm$^{-2}$).

$$E(\nu) = \int \sigma(\nu)\Lambda(\nu)d\nu \cdot n \cdot V \cdot \Phi \tag{3}$$

We excite NO near 214.8800 nm (46537.64 cm$^{-1}$) at an envelope with 4 overlapping rotational lines (Q-branch $J'' = 2.5; 3.5$ and P-branch $J'' = 1.5; 2.5$). The peak of the absorption cross section in this envelope at 300K is $1.5 \times 10^{-16}$ cm$^2$ molecule$^{-1}$.

Because the laser used here has a linewidth that is comparable to the Doppler broadened linewidth of NO at 300K ($\Delta\nu_{Doppler} = 3$ GHz) we approximate the convolution of the NO cross section with the laser lineshape as $\int \sigma(\nu)\Lambda(\nu)d\nu \approx \sigma = 1.5 \times 10^{-16}$ cm$^2$ molecule$^{-1}$. In reality, the non-negligible laser lineshape in the current system somewhat reduces the effective $\sigma$. A typical cell pressure that has been used in the laboratory and can be maintained during aircraft sampling up into the lower stratosphere is 42.5 hPa ($1.05 \times 10^{18}$ molecules cm$^{-3}$). Therefore, at a NO mixing ratio of 1 pptv the NO concentration in the cell would

be $n = 1.05 \times 10^6$ molecules cm$^{-3}$. We estimate that a cubic volume with an edge of about 5 mm is imaged onto the PMT. With a typical laser power of 1 mW at 215 nm the photon flux is $\Phi = 4.4 \times 10^{15}$ photons s$^{-1}$ cm$^{-2}$. Therefore the estimated excitation rate is 86,625 s$^{-1}$ pptv$^{-1}$ mW$^{-1}$.

The fluorescence quantum yield $\phi$ is determined by the competition between the natural fluorescence lifetime of NO* and collisional quenching by other molecules in the sample gas. The natural radiative lifetime of $A^2\Sigma(v' = 1)$ is 200 ns

($k_r = 5 \times 10^6$ s$^{-1}$) (Luque and Crosley, 2000). The primary quenchers in the atmosphere are expected to be N$_2$, O$_2$ and Ar. Nee et al. (2004) measured the quenching rate coefficients for NO $A^2\Sigma(v' = 1)$ by N$_2$, O$_2$, and Ar to be $6.1 \times 10^{-13}$, $1.48 \times 10^{-10}$, and $3.3 \times 10^{-13}$ cm$^3$ molecules$^{-1}$ s$^{-1}$. Therefore in dry air with 78% N$_2$, 21% O$_2$ and 1% Ar, the quenching rate is $k_{air} = 3.2 \times 10^{-11}$ s$^{-1}$. Carbon dioxide is also a fast quencher of NO*, ($k_q = 3.8 \times 10^{-10}$) (Nee et al., 2004), and 400 ppm of CO$_2$ would increase the fluorescence quenching rate by 0.4%. For measurements of NO near very large CO$_2$ sources

this additional quenching might need to be considered. Water vapor also has an important and variable effect on $\phi$ and this is addressed in a subsequent section. In 42.5 hPa of dry air, we therefore expect the fluorescence quantum yield to be:

$$\phi = \frac{k_r}{k_r + k_{air}[M]} = 0.13 \tag{4}$$

Similarly, the $e$-folding lifetime of the fluorescence signal in these conditions is calculated to be 26 ns. By adjusting the photon counting gates in our system in 5 ns increments, we determined the signal lifetime at a range of pressures from 14.7 hPa to

103.5 hPa. A Stern-Volmer analysis of the observed lifetimes concluded that the natural radiative lifetime $\tau_r$ is 180 ns, and the quenching rate in dry zero air (no CO$_2$) is $3.6 \times 10^{-11}$ cm$^3$ molecules$^{-1}$ s$^{-1}$, overall in good agreement with the literature values of 200 ns and $3.2 \times 10^{-11}$ cm$^3$ molecules$^{-1}$ s$^{-1}$.

The fluorescence collection efficiency is determined by the product of the optical bandpass filter transmission with the geometric collection efficiency and the quantum efficiency of the PMT. The Franck-Condon factor for the $A^2\Sigma(v' = 1) \rightarrow$

$X^2\Pi(v'' = 4)$ transition is 0.13 (Danielak et al., 1997) and the bandpass filter transmission at 255 nm is 0.63, while the Franck-Condon factor for $A^2\Sigma(v' = 1) \rightarrow X^2\Pi(v'' = 5)$ is also 0.13 (Danielak et al., 1997) and the bandpass filter transmission at 267 nm is 0.69. The 0.5 NA lens captures 0.067 of the fluorescence emitted from the center of the cell. At best, a mirror opposite the lens increases this collection by a factor of 1.5, bringing the geometric collection efficiency to 0.1. The quantum efficiency





of the PMT at 255 - 267 nm is $\sim 0.2$. Thus, the fluorescence collection efficiency of the system is $\Omega = (0.13 \cdot 0.63 + 0.13 \cdot$
$0.69) \cdot 0.1 \cdot 0.2 = 3.4 \times 10^{-3}$.

Taking the product of the excitation rate with the fluorescence quantum yield and fluorescence collection efficiency, we esti-
mate that the anticipated signal rate is approximately 38 counts $s^{-1}$ $pptv^{-1}$ $mW^{-1}$. The anticipated signal level is comparable
to the fluorescence sensitivity that has been measured for this instrument (11.3 counts $s^{-1}$ $pptv^{-1}$ $mW^{-1}$ see Figure 10). The
comparison between the theoretical sensitivity and measured sensitivity is quite reasonable given uncertainties in a number of
the parameters discussed above.

## 2.4 Temperature, pressure and water vapor dependence

The absorption cross-section is proportional to the rotational populations in the ground states that are being probed ($J'' =$
1.5; 2.5; 3.5) and, therefore, a temperature dependence to the signal is anticipated. Near 300K, populations in all of the probed
rotational states will decrease with increasing temperatures. Figure 5 shows the calculated temperature dependence of the
Doppler broadened absorption cross-section relative to 300K at 214.880 nm. In this region, a relative decrease in sensitivity of
0.34% $K^{-1}$ is calculated.

If changes in the sample temperature were identical to changes in the temperature of the gas in the reference cell, any
sensitivity changes would be exactly accounted for during data reduction. If significant differences arise in the gas temperatures
between the sample and reference cells, an artifact would arise. For ground based measurements, the temperatures of the
sample cell, reference cell and sample gas flow will usually be very similar. For aircraft measurements where gas from the cold
atmosphere (as low as -90K) is rapidly drawn into a warmer analysis region, care must be taken to ensure the probed sample
gas is well thermalized with the measurement cell and that the reference cell is close in temperature.

The pressure dependence of the signal arises due to changes in both the excitation rate $E(\nu)$ and the fluorescence quantum
yield $\phi$. The excitation rate is directly proportional to the NO concentration in the cell and will increase linearly with pressure
at a constant NO mixing ratio. In the low pressure limit, $\phi$ is independent of pressure, and in the high pressure limit, $\phi$ is
inversely proportional to the pressure. Thus, at low pressures the LIF signal increases with pressure and eventually the signal
becomes independent of pressure.

Figure 6 shows the calculated pressure dependence of the LIF signal based on the previously cited fluorescence and quench-
ing rates. We measured the LIF signal between 20 and 100 hPa and show that it generally follows the anticipated pressure
dependance. At 100 hPa, we observed $\approx 10\%$ more signal than we do at 30 hPa. Significantly higher pressures reduce the
instrumental response time and cannot be maintained on aircraft in the upper troposphere with sufficient flow.

Quenching of $NO^*$ due to water vapor is fast and must be considered in humid environments. While a quenching rate
coefficient ($k_{H2O}$) of the $A^2\Sigma(v' = 1)$ state has not been reported, Paul et al. (1996) reported for quenching of the $A^2\Sigma(v' =$
0) state $k_{H2O} = 8.97 \times 10^{-10}$ $cm^3$ molecules$^{-1}$ $s^{-1}$. Based on the very small differences between quenching rates for the
$A^2\Sigma(v' = 1)$ and $A^2\Sigma(v' = 0)$ states by $N_2$ and $O_2$ (Nee et al., 2004), it was expected that quenching by $H_2O$ of $A^2\Sigma(v' = 1)$
is reasonably close to the value measured for $A^2\Sigma(v' = 0)$ and therefore would be important for high humidities.




Quenching by $H_2O$ will decrease the sensitivity of the instrument to NO by introducing an additional term in the fluorescence quantum efficiency:

$$\phi = \frac{k_r}{k_r + k_{air}[M] + k_{H2O}[H_2O]} \tag{5}$$

Defining $\phi_0$ as the fluorescence quantum efficiency at zero $H_2O$ concentration, it follows that:

$$\frac{\phi_0}{\phi} = 1 + \frac{k_{H2O}}{k_r + k_{air}[M]}[H_2O] \tag{6}$$

Therefore, a plot of the inverse of the relative LIF signal ($S_0/S$) as a function of $[H_2O]$ will yield a line with a slope equal to $\frac{k_{H2O}}{k_r + k_{air}[M]}$. For cell pressures of 42.5 hPa and 85.4 hPa we measured $S_0/S$ for a range of $H_2O$ mixing ratios and used this analysis to determine $k_{H2O}$. For these experiments, mixtures of 5 ppb NO in zero air under a range of humidities was generated using varying mixtures of saturated zero air and dry zero air. The mixture was sampled in parallel by the LIF instrument and an MBW 373LX chilled mirror hygrometer (MBW Calibration AG). Using this analysis and assuming $k_r = 5 \times 10^6$ s$^{-1}$ and $k_{air} = 3.2 \times 10^{-11}$ cm$^3$ molecules$^{-1}$ s$^{-1}$, the data from 42.5 hPa and from 85.4 hPa suggest that $k_{H2O}$ is $6.4 \times 10^{-10}$ cm$^3$ molecules$^{-1}$ s$^{-1}$ and $7.4 \times 10^{-10}$ cm$^3$ molecules$^{-1}$ s$^{-1}$, respectively. The discrepancy between these results could arise from small errors in any of the parameters $k_r$, $k_{air}$, $[M]$ or $[H_2O]$ used to derive these values. In Figure 7 we show the observed relative signals as a function of $H_2O$ at the two cell pressures. A mean value of $k_{H2O}$ of $6.9 \times 10^{-10}$ cm$^3$ molecules$^{-1}$ s$^{-1}$ is used to reasonably reproduce the observations at both cell pressures.

Figure 7 shows that the decrease in the LIF sensitivity to NO under humid conditions is quite substantial (e.g. a 29% decrease in signal at 20,000 ppm $H_2O$). However, this change in instrument sensitivity is well understood and can be characterized with high accuracy for a known or constant LIF cell pressure. The bottom panel of 7 shows that at up to 20,000 ppm $H_2O$, differences of up to 3% are observed between the laboratory observations and the model. We note that for such a $H_2O$-dependent sensitivity to be applied during data reduction, good quality measurements of $H_2O$ mixing ratios are required and the uncertainty in the $H_2O$ measurements will contribute to the uncertainty in the calculated NO mixing ratio.

To test for the ability of in-situ $H_2O$ measurements to account for the reduced fluorescence quantum yield, we performed standard addition calibrations of NO into both ambient and dry zero air during DC-8 flights for the FIREX-AQ experiment. A typical calibration sequence involved a 30 second period of adding 5 sccm of 5 ppmv NO into the normal sample flow of 2500 sccm, followed by 30 seconds where the ambient flow was also displaced using dry zero air. For each pair of measurements, the ambient water vapor mixing ratio was determined using a Los Gatos Research analyzer for $N_2O$, CO and $H_2O$ (LGR). Data from those calibrations are shown in Figure 7. The effect of $H_2O$ on the NO signal as determined both in laboratory and on the DC-8 agree quite well, with maximum differences near 2%. At $H_2O$ mixing ratios greater than $\sim$ 15,000 ppmv, the differences between the model and in-situ observations diverge somewhat, with maximum differences of 4%.

## 3 Operation

Figure 8 shows the fluorescence signal that is typically observed when scanning the seed laser current in the region used for NO measurements. We show two equivalent laser scans on a sample of 10 ppbv NO calibration gas. Each scan shows about 15





s of data acquisition. The observed fluorescence spectrum is plotted on top of a theoretical absorption spectrum that has been
calculated using the PGOPHER program at a temperature of 300K. The wide wings of the observed spectral lines are believed
to be due to spectral broadening in the fiber laser system, most likely due to self-phase modulation. In the future it should be
possible to reduce this broadening which would both increase the online signal somewhat and reduce the online/offline tuning
separation required.

Here we use the feature near 214.88 nm as the NO 'online' signal and a minimum in the fluorescence near 214.89 nm as the
NO 'offline.' For ambient measurements, we typically tune the laser online for 80 ms, followed by a 20 ms measurement of the
offline signal. Figure 9 shows a typical five second segment of ambient measurement data. A small amount of hysteresis when
tuning online/offline is sometimes apparent due to the finite impedance of the seed laser diode/driver system. However, these
transients are also captured in the reference cell signal and therefore do not contribute to increased uncertainty in the calculated
NO mixing ratio.

The reference cell is used primarily to maintain the laser wavelength near the peak of the NO online feature. This is ac-
complished by continuously walking the laser wavelength around the peak (typically with a period of $\sim 10$ s) to maintain a
local maximum signal in the reference cell. A fixed differential seed laser current is maintained between the online and offline
positions. For data reduction, a running average of the offline signal (counts $\mathrm{mW}^{-1}$) is subtracted from the online signal in
both the measurement and reference cells. Then, the online/offline difference in the measurement cell is normalized to the on-
line/offline difference in the reference cell. This normalization accounts for the known small changes in instrument sensitivity
when walking off the side of the NO peak, as well as any unintentional changes such as pressure fluctuations, or small changes
in the laser linewidth.

## 4    Linearity

The dynamic range of the instrument is limited by the pulse-pair resolution of the photon counting system (20 ns) and the
repetition rate of the laser (320 kHz). At 85 hPa cell pressure, the lifetime of the fluorescence signal is primarily controlled
by quenching and is 14 ns. Since the pulse-pair resolution is greater than the signal lifetime the system will at most count one
fluorescent photon per laser shot and therefore at very high signal levels the observed count rate will start to deviate from a
linear response to the rate at which photons strike the photocathode. However, as discussed previously (Wennberg et al., 1994;
Rollins et al., 2016) under the conditions of a known maximum count rate (320 kHz) the observed count rate can be corrected
exactly to match the true signal rate and thereby significantly increase the dynamic range without loss of accuracy. At the
present typical signal rates ($\sim 10$ counts $\mathrm{s}^{-1}$ $\mathrm{pptv}^{-1}$), errors associated with saturation would not be encountered for NO less
than 100 ppbv. Figure 10 shows a typical calibration slope measured during a flight by dynamic dilution of an NO standard
into the instrument inlet.



## 5   Photolytic interferences

Photolysis of other species to produce NO within the LIF cell by the probe laser could in principle pose an interference. Species known to photolyze at 215 nm producing NO include $NO_2$ ($\sigma = 5.0 \times 10^{-19}$ cm$^2$ molecule$^{-1}$), HONO ($\sigma = 1.9 \times 10^{-18}$ cm$^2$ molecule$^{-1}$), and ClNO ($\sigma = 1.6 \times 10^{-17}$ cm$^2$ molecule$^{-1}$). Using the estimated photon flux of $4.4 \times 10^{15}$ photons s$^{-1}$ cm$^{-2}$ we calculate for the species with the largest absorption cross-section (ClNO) the photolysis rate in the probe volume of the LIF cell would be approximately 0.07 s$^{-1}$. We estimate that the residence time in this volume is less than 0.01 s and therefore that

less than 0.07% of any sampled ClNO could be photolytically converted into NO. Photolysis conversion for HONO and $NO_2$ would be respectively 10-100 times smaller. Such interferences are therefore negligible considering typical concentrations of these other species relative to NO in the atmosphere. A lack of significant photolytic interferences is confirmed by some of the nighttime FIREX-AQ observations when the LIF measurements show NO < 0.1% of the simultaneously measured $NO_2$ on the aircraft.

## 6   Accuracy

Calibration is accomplished periodically during operation by adding typically 2-10 sccm of a 5 ppmv NO in $N_2$ mixture to the instrument sample flow of 2500 sccm. This provides calibration mixing ratios of 4-20 ppbv. Typically, the analytical accuracy of the NIST traceable NO standard is $\pm 1\%$. The flow controller that delivers that NO to the inlet and the mass flow meter which measures total inlet flow are routinely checked against Bios Drycals with $\pm 1\%$ uncertainty each. Addition of

these uncertainties in quadrature suggests that the uncertainty in the mixing ratio of NO delivered to the inlet for a single calibration point is $\pm 2\%$. For airborne measurements where water vapor mixing ratios may change rapidly and high accuracy water vapor measurements are available to correct for the NO fluorescence quantum yield, the instrument is calibrated in zero air by additionally overflowing the inlet with dry zero air (<10 ppmv water vapor).

During FIREX-AQ, the NO-LIF instrument logged more than 140 hours of airborne operation over 22 flights. Typically,

calibrations were performed once per hour during flights. Throughout the mission, a range of calibration coefficients were measured spanning a $\pm 5\%$ range from the mean calibration coefficient, while the precision of each measurement can explain less than $\pm 1\%$ of this variation. These variations in the measured calibration factors were more than was typically observed during laboratory operation, and although the calibration coefficients do not systematically vary with the environmental temperature, it is believed that the instability in sensitivity is related to the temperature of the DC-8 cabin, which varied from

roughly 25 - 40°C. This range may be due to apparent sensitivity changes related to temperature effects on the flow/calibration system, or real changes in sensitivity due to perhaps an optical effect (e.g., etalon). The cause for the range in calibration coefficients measured in flight will be a focus of future investigation. For now, we conservatively add in quadrature this $\pm 5\%$ uncertainty to the $\pm 2\%$ uncertainty in the mixing ratio delivered during a calibration to arrive at a $\pm 6\%$ uncertainty in the sensitivity of the instrument in dry air.

For humidity corrections, we need to consider both the uncertainty in the water vapor measurement and in the model used to calculate the correction (i.e., Fig. 7). Typically, water vapor measurements from aircraft are known to $\pm 10\%$. By applying





a 10% uncertainty to the relationship shown in the top panel of Fig. 7 for 85.4 hPa cell pressure, we derive an $H_2O$ dependent uncertainty associated only with the uncertainty in the water vapor measurement. This relationship is roughly linear with 0% additional uncertainty in dry air, and 3% uncertainty at 20,000 ppm $H_2O$. For $H_2O$ greater than 10,000 ppmv, deviations

between the modeled and measured effect of $H_2O$ on the fluorescence quantum yield (bottom panel Fig. 7) would add an additional 2-3% uncertainty. This uncertainty can likely be reduced in the future by improvements in the model used to reproduce the observed effect of $H_2O$ on the NO-LIF signal.

## 6.1 Detection Limit

The precision with very low mixing ratios of NO in the system was measured in the laboratory to test for any zero artifacts and

to determine the instrumental detection limit. For these tests, the observed data were analyzed assuming that no artifact of any kind exists (i.e., the sampled NO mixing ratio is proportional to the difference between online and offline signals at all mixing ratios), and the instrumental response as determined by additions of NO standards is linear down to zero concentration. Doing this, we typically find that flowing air directly from zero air cylinders (Praxair) into the instrument results in a measurement of 1-2 pptv NO. The observed NO was reduced to less than 0.2 pptv by flowing zero air through a potassium permanganate trap

($KMnO_4$).

Figure 11 shows the distribution of 1 Hz NO mixing ratios measured when sampling $KMnO_4$ scrubbed zero air for 1.4 hours in the laboratory. During this period, the mean NO measured was 0.19 pptv and the noise was normally distributed with a $2\sigma$ width of 0.76 pptv. For this period, the laser power was 0.9 mW, and the NO sensitivity was 10 counts $s^{-1}$ $mW^{-1}$. The average count rate was 12 counts $s^{-1}$, and thus the calculated background count rate is 10.3 counts $s^{-1}$. The width of the

observed mixing ratio distribution is what would be expected from a Poisson limited distribution of the photon counts ($\sigma = \sqrt{(12 \text{ counts})}/(10 \text{ counts mW}^{-1} \text{ pptv}^{-1} \times 0.9 \text{ mW}) = 0.385$ pptv). This suggests that no sources other than photon counting statistics contribute significantly to the precision near the detection limit. The calculated $2\sigma$ detection limit for a 1 second integration is therefore 0.97 pptv, and for ten seconds is 0.25 pptv. No evidence exists to suggest that the 0.19 pptv observed in the scrubbed zero air is due to anything other than NO remaining in that sample.

Figure 12 shows an Allan deviation analysis of the scrubbed zero-air sampling. During data reduction, a choice must made about what duration to use for averaging of the offline signal. Sufficiently long averaging effectively eliminates the offline signal as a source of noise, while shorter averaging assures that any changes in offline signal are completely resolved. To illustrate this two analyses are shown, one where a 1 hour average is used for the offline and another using a 1 second average offline. At integration times of less than 100 seconds, the analysis using a 1 hr average of the offline signal shows that the precision is

limited only by the counting statistics associated with the online signal. Instabilities in the offline signal with time constant on the order of 100-1000 seconds leads to lower $\sigma$ using the 1 s offline average for integrations exceeding $\sim$ 10 minutes.





## 7 In-situ results

FIREX-AQ provided an excellent opportunity for comparing the LIF instrument to a state-of-the-art CL instrument. The CL instrument was located at the front of the cabin, and sampled from a probe on the port side of the aircraft. The LIF instrument
was located mid-cabin, with a probe extending from the starboard side of the aircraft. The LIF instrument shared the probe described by Cazorla et al. (2015) with four other instruments, each of which sampled about 2 slpm from the total flow of more than 20 slpm. Table 1 compares key performance and physical characteristics for the LIF and CL instruments.

In Figure 13, we show a time series of the 1 Hz measurements from a 3.4 hour flight on July 24, 2019. This flight shows typical results from the mission. Generally, agreement between the LIF and CL instruments is excellent and we have no
evidence of detectable interferences for either instrument. Small differences were sometimes observed when leaving large plumes where the NO mixing ratio would decrease by more than one order of magnitude over the period of one second. These are believed to be due to a volume in the CL instrument sample line, which is designed to match the $NO_2$ photolysis volume in a paired channel. The lower noise of the LIF instrument is apparent primarily at mixing ratios lower than 10 pptv. Figure 14 shows scatter plots of the LIF and CL data for two flights. The top panel shows the comparison from a flight on July 22, 2019
during which the DC-8 sampled air throughout the California San Joaquin Valley, the Los Angeles Basin, and then transited at 12.5 km altitude to Boise, ID. The bottom panel shows measurements from the flight on July 25, 2019 where the DC-8 sampled wildfire smoke while based in Boise, ID. In both figures, the data are colored by the water vapor mixing ratio measured by LGR to demonstrate that once the data are adjusted for the measured water vapor measurements, systematic differences due to differences in water vapor are not apparent. For all data shown in Figure 14, the regression fit slope is 0.993, indicating that the
LIF measured NO was on average 0.7% lower than CL - a difference easily attributable to calibration uncertainties for either instrument.

## 8 Conclusions

A new instrument has been described for performing direct measurements of atmospheric NO using single-photon laser-induced fluorescence. The demonstrated detection limit for 10 s of integration is 0.3 pptv, and to our knowledge this is the
lowest detection limit at this time resolution that has been demonstrated for an airborne atmospheric NO sensor. Besides having excellent precision, the instrument has significant practical advantages as compared to CL instruments. Consumables such as dry ice and pure oxygen are not required. CL instruments have background levels on the order of 10-100 pptv equivalent and the background typically decreases for a number of hours after instrument operation begins. CL background also increases significantly at high altitudes and latitudes due to the effect of cosmic rays on the large infrared-sensitive PMTs. These issues
mean that at higher altitudes CL instrumental precision will be degraded. This can clearly be observed in Figure 13 where the CL precision degraded significantly relative to LIF as the DC-8 climbed from 2 km to 10 km altitude. The variable background in CL also means that for accurate measurements on the order of 10 pptv to be made, frequent zero determinations must be performed, and running the instrument for a number of hours before measurements are made is desirable. For this reason, the





LIF instrument requires less effort to operate and has the potential to be more accurate at low mixing ratios for typical aircraft
experiments where continuous running prior to flights adds an additional experimental burden.

The LIF instrument performed well without failure and without a dedicated operator for the 22 science flights during the
NASA/NOAA FIREX-AQ mission. Precision in flight was typically not as good as demonstrated in the laboratory, and this
was due to reductions in laser power associated with the wide range of cabin air temperatures ($\sim$ 25 - 40°C) experienced on
the DC-8. Future improvements in the thermal management of the laser system are expected to improve this issue. In addition,
a number of possibilities exist to further improve the signal level. These include: laser linewidth reduction, improved bandpass
filter to collect the $v'' = 3$ emission and increase the transmission at $v'' = 4$ and 5, increased laser power, and increases in the
geometric fluorescence collection efficiency. Altogether, we estimate that increases in signal by as much as a factor of 10 are
possible.

The one notable additional challenge associated with this LIF technique is that the signal is significantly reduced in the
presence of high water vapor mixing ratios. Therefore, a fast and accurate water vapor measurement must be deployed with
the NO-LIF instrument to obtain accurate NO measurements in the planetary boundary layer from aircraft. For ground-based
operations where water vapor changes much more slowly, it may be acceptable to periodically calibrate the sensitivity in
ambient air. For measurements in the upper troposphere and stratosphere where $H_2O$ mixing ratios are generally less than 1000
ppmv, this effect is negligible. We showed that use of the LGR measurements on the DC-8 allow us to remove any observable
difference between the CL and LIF techniques associated with variable water vapor. A potential alternative strategy in the
future is to use a mixture of ambient air doped with high mixing ratios of NO in the reference cell, instead of the zero air / NO
reference mixture.

The new sensor has the potential to provide high confidence in future measurements of atmospheric NO at mixing ratios of
less than 10 pptv which are characteristic of much of the global remote marine boundary layer (Singh et al., 1996; Bradshaw
et al., 2000). The technique can be extended to perform measurements of $NO_2$ using selective photolytic conversion to NO
(Pollack et al., 2010), or total reactive nitrogen using catalytic conversion (Kliner et al., 1997; Ryerson et al., 1999). In either
case, the LIF instrument could be operated with a significantly reduced flow rate to enable the use of a smaller converter than
what is typically required for use with a CL based NO detector. In addition, we have demonstrated the ability to make an
isotopologue specific measurement, and have observed in the laboratory signals also from the $^{15}N^{16}O$ isotopologue. Future
efforts will focus on quantifying $^{15}N/^{14}N$ and $^{18}O/^{16}O$ ratios which are unique tools for identifying sources of atmospheric
$NO_x$ and diagnosing atmospheric oxidation chemistry.

*Data availability.*   The data collected on the DC-8 are available on the NASA/NOAA FIREX-AQ data archive: https://www-air.larc.nasa.gov/cgi-
bin/ArcView/firexaq.



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





**Table 1.** Specifications of CL and LIF instruments compared on the NASA DC-8 during FIREX-AQ. The LIF instrument is a two channel detector, while the CL instrument has four channels, including NO, $NO_2$, $NO_y$ and $O_3$. Therefore, comparisons of weight and power of the instruments should consider these differences.

|  | LIF | CL |
| --- | --- | --- |
| Sensitivity | 10 CPS / pptv | 10 CPS / pptv |
| Background | 10 CPS | 800-1100 CPS |
| Detection Limit (1 Hz, $2\sigma$) | 1 pptv | 6 pptv |
| Consumables | trace NO for ref. cell | pure $O_2$, cryogen |
| Power consumption | 400 W | 2100 W |
| Mass | 50 kg | 150 kg |
| Quenching by 10,000 ppm $H_2O$ | 16% | 4% |





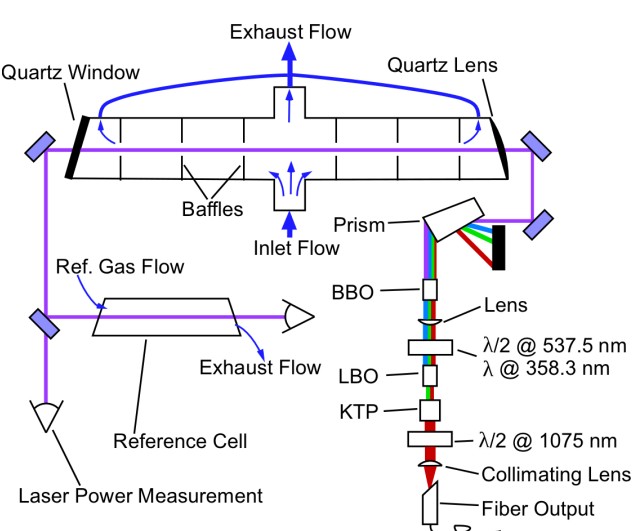

**Figure 1.** Schematic of the free-space optical layout.

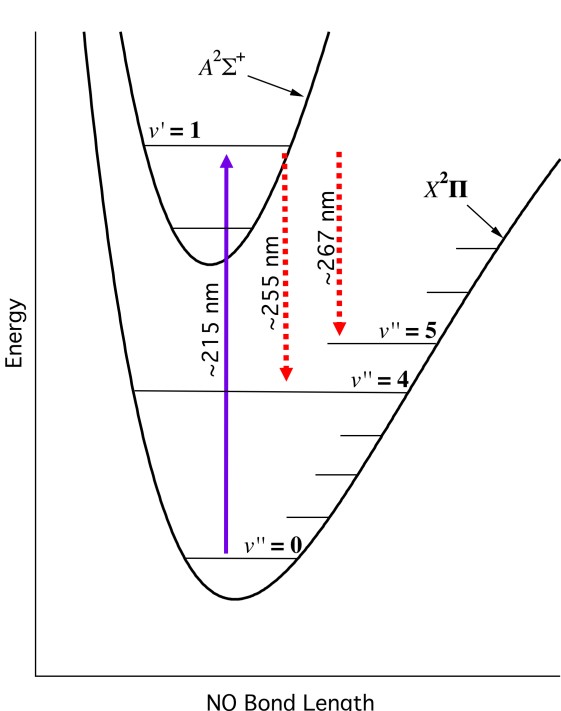

**Figure 2.** Energy level schematic of NO-LIF system.



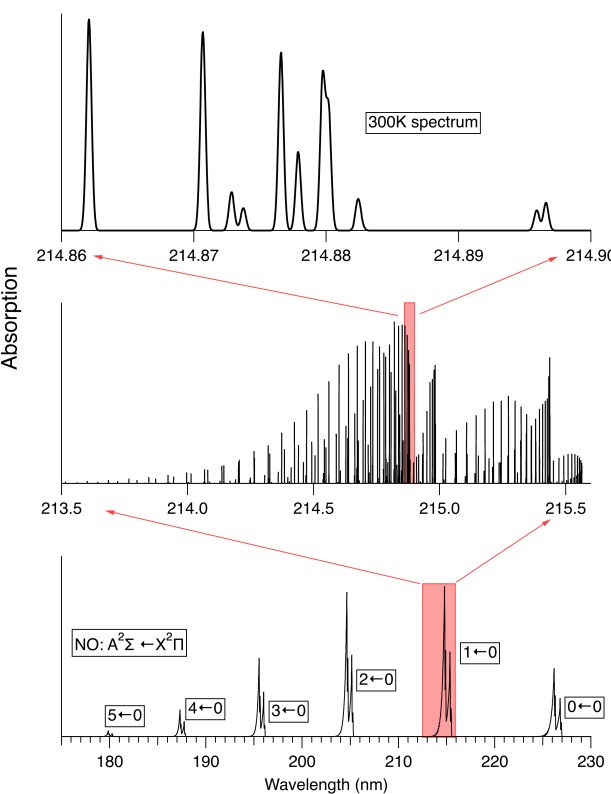

**Figure 3.** Spectrum of the $A^2\Sigma \leftarrow X^2\Pi$ transition in NO at 300 K. Bottom panel shows the absorption spectrum in low resolution. The middle panel shows the calculated stick spectrum for the $1 \leftarrow 0$ transition. The top panel shows the spectral region used here to measure NO, convolved with a $0.1\ \text{cm}^{-1}$ width Gaussian lineshape, which is the Doppler linewidth of NO at 300K.





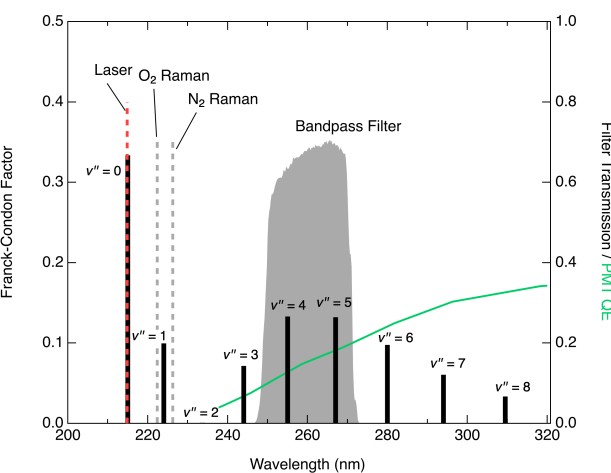

**Figure 4.** Franck-Condon factors (black sticks plotted on left axis) indicating the expected distribution of the fluorescence intensity from the $A^2\Sigma(v'=1) \to X^2\Pi$ relaxation. Grey shaded region (right axis) shows detected spectral region in this work. Locations of significant scatter from Rayleigh, $O_2$ Raman and $N_2$ Raman are also indicated. Green line plotted against the right axis shows the quantum efficiency (QE) of the PMT module used in this work


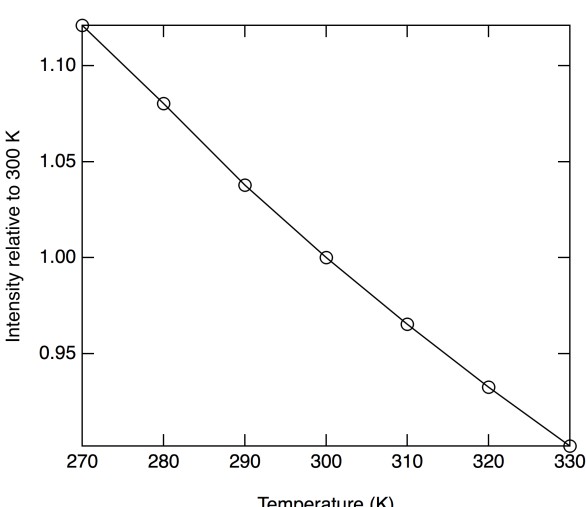

**Figure 5.** Calculated temperature dependance of the absorption cross section near 300K.



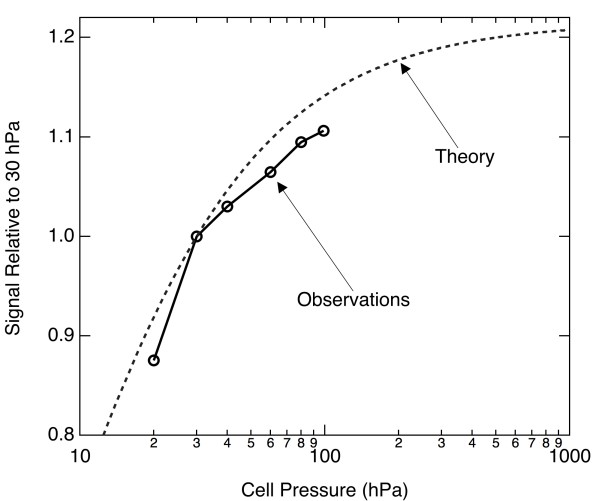

**Figure 6.** Calculated and measured dependence of the LIF signal on sample cell pressure.



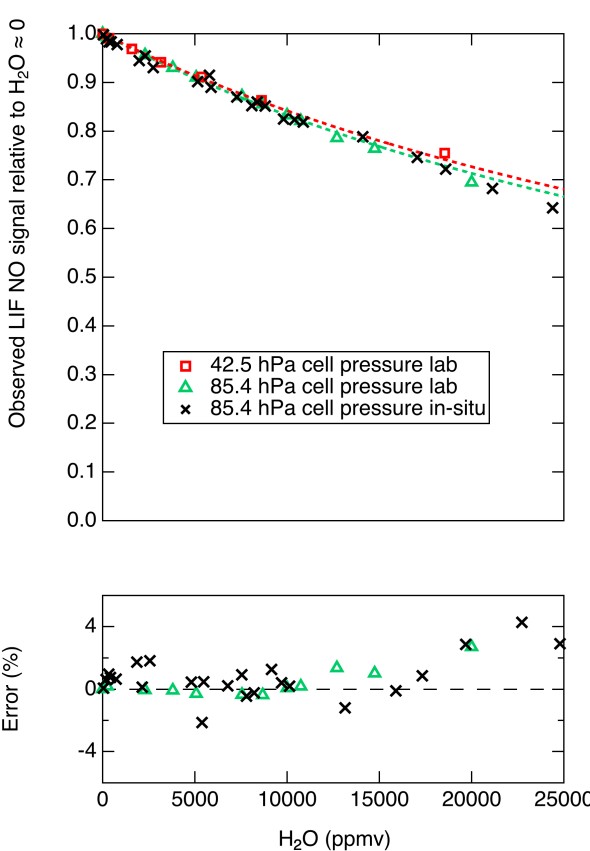

**Figure 7.** Top panel: Markers show the observed $S/S_0$ values, and dashed lines show the calculated $S/_0S$ values using $k_{H2O} = 6.9 \times 10^{-10}$ cm$^3$ molecules$^{-1}$ s$^{-1}$. Bottom panel: Relative difference between observed and calculated $S/S_0$ values shown in the top panel.



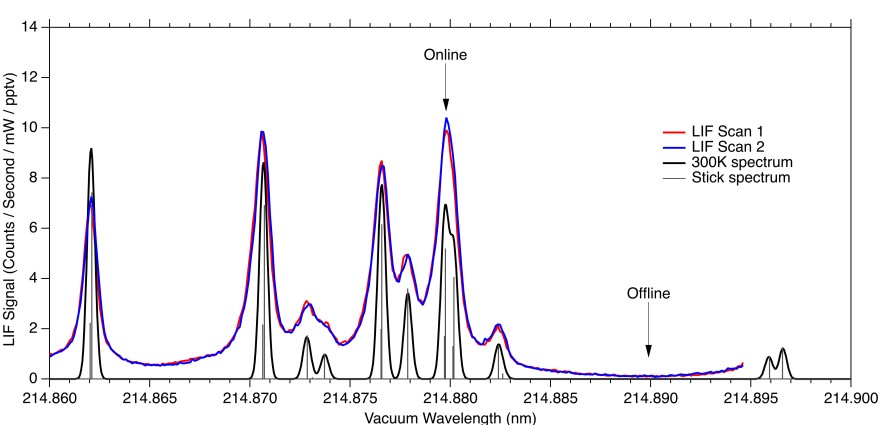

**Figure 8.** LIF signal observed during laser scans compared to calculated absorption cross section at 300K.



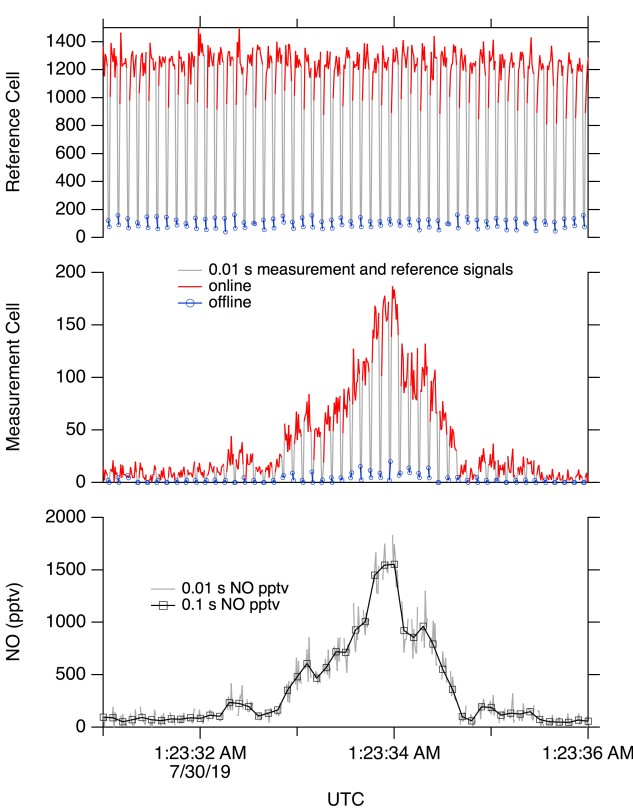

**Figure 9.** A five second segment of typical data. Bottom panel shows observed online and offline signal rates. Top panel shows the calculated NO at 10 Hz.





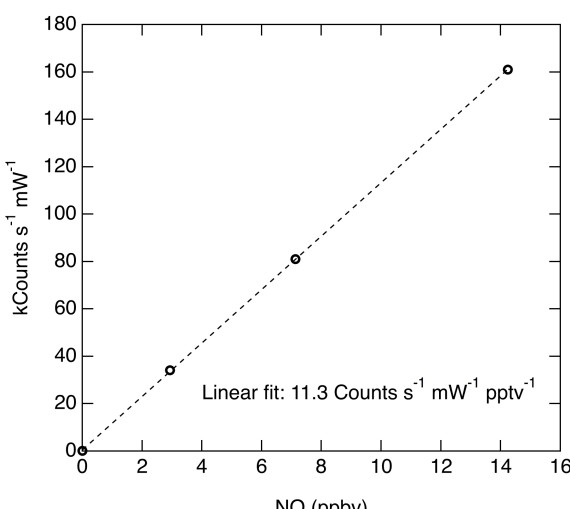

**Figure 10.** Typical calibration data acquired during a flight when sampling a mixture of NO in zero air.



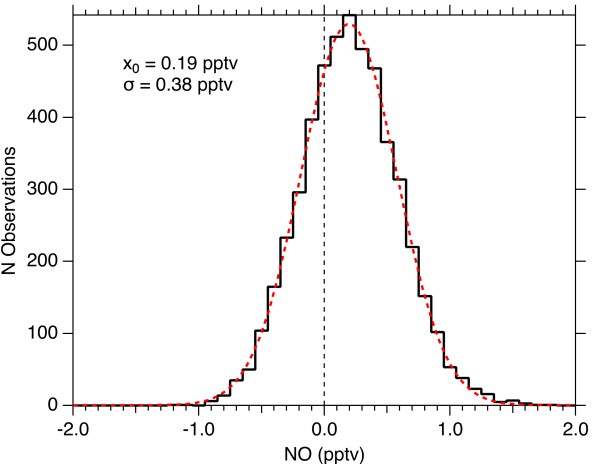

**Figure 11.** Histogram of observations sampling zero air scrubbed with potassium permanganate



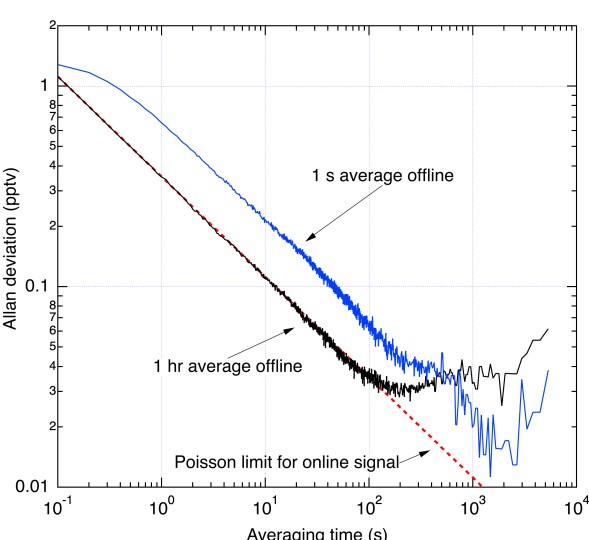

**Figure 12.** Allan deviation analysis from 1.4 hours of sampling scrubbed zero-air in the laboratory.



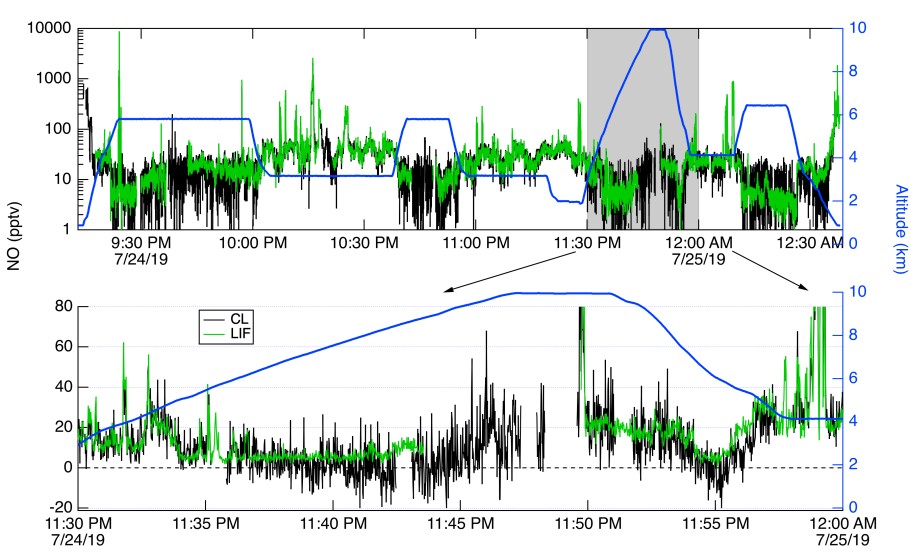

**Figure 13.** Timeseries of CL (black) and LIF (green) data from a DC-8 flight on July 24, 2019. The top panel shows all of the flight, with the altitude of the DC-8 plotted on the right axis in blue. The bottom panel shows a 20 minute segment expanded.

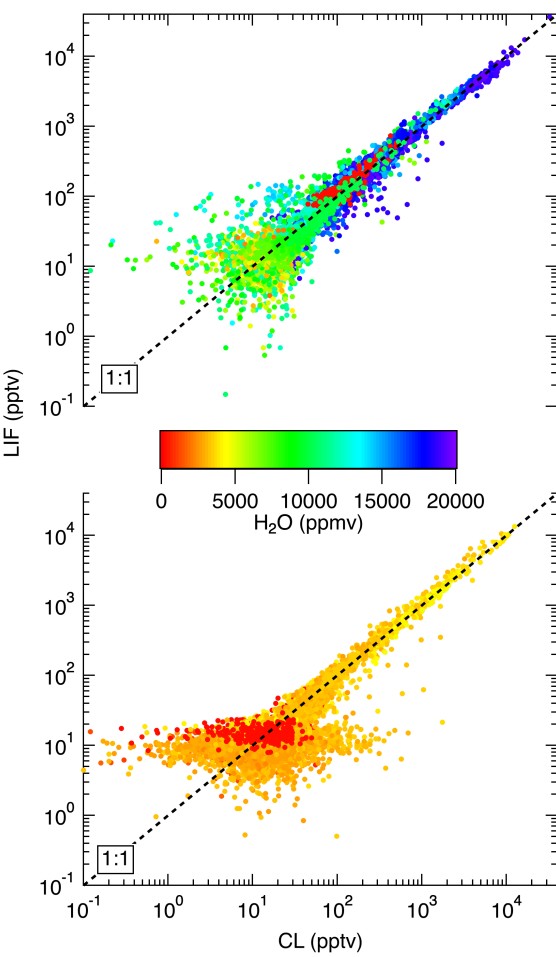

**Figure 14.** Comparison between LIF and CL measurements from the NASA DC-8 during FIREX-AQ. Top panel shows a comparison of the 1-second measurements from the flight on July 22. This flight sampled air from the LA basin, San Joaquin Valley and from the free troposphere transiting from Palmdale, CA to Boise, ID. Bottom panel compares data from a flight on July 25 focusing on wildfire smoke sampling. Differences in LIF precision between these flights are due to unusually low laser power on the July 22 flight.