# Peer review of "Single-photon laser-induced fluorescence detection of nitric oxide at sub-parts per trillion mixing ratios"

_Atmospheric Measurement Techniques, 2020_

## Referee Comment (RC1) · Anonymous Referee #1 · 2 Mar 2020

This paper describes an LIF-based instrument for airborne observations of NO in the troposphere and stratosphere. Details are provided on measurement theory and performance characteristics. It is shown that this instrument performs as well, or better than, a state-of-the-art Cl-based instrument. The paper is well-written and the number and style of figures is appropriate. My comments are relatively minor, and publication is recommended.

L23: while it may be true that very low NO chemistry remains poorly understood, it is known that many of the OH observations used in the Rohrer 2014 study suffered from positive artifacts. So, not sure if this is the best reference. Perhaps there are more

recent observations that are artifact-free and illustrate this point (e.g. from SOAS or GOAMAZON)?

L64: "repetition"

L178: What is the actual width of the fluorescence collection gate (in ns) used in data acquisition? Does it exclude the laser pulse?

L206: -90K is very cold indeed!

L215: "dependence"

L268: what is the width of the running average? And is it just a boxcar window?

Sect. 7: Is there any significant background variability in the FIREX data beyond what is seen in the lab? If so, it seems like this would affect the chosen background smoothing window.

L373 – 378: These statements would fit better in Section 7.

L404: Bringing up the isotopologue detection here seems out-of-place. While this is exciting, it might be better to state that this is possible rather than to state that you have done it (unless you want to show some data to support it).

Figure 9: Caption incorrect.

---

## Referee Comment (RC2) · Anonymous Referee #2 · 3 Mar 2020

General comment:

The manuscript is suitable for publication in the AMT journal. Overall, it is well written and provides important information about the features (supported by theory and observations) of a laser-induced fluorescence system for the direct detection of atmospheric nitric oxide. I recommend a publication after minor revision according to the following comments.

Specific comments:

Line (1-8): Please provide key numbers/values (e.g., wavelength, uncertainty, etc.) in the abstract.

Line (37): What is the function of dry ice in a CL instrument? Please can you elaborate or provide a reference?

Line (44-45): How do you know it is red-shifted (Measured or from literature)?

Line (63): Why is it important to resolve Doppler broadened in the case of the LIF technique?

Line (66): Is the detection limit at 10 seconds measured or calculated by simply dividing 1 ppt with sqrt (10)? How do you know it follows the square root dependency?

Line (79-82): The sentences "The DFB output…...an end capped fiber" are not easily understandable. Describe the purpose, why is this performed or what is required to achieve? Please modify it for simplicity.

Line (94): What is the purpose of the reference cell? Why is it necessary to have 500 ppb of NO? This needs a further explanation for a reader.

Line (96): What is the absolute pressure? Is there an orifice involves or just pump is used to achieve a lower pressure inside both cells?

Line (97-101): It is not clear from Figure 1 the location of the fused silica lens and the bandpass filter? What type of signal comes from the reference cell (I assume absorption)? Which of the PMT in Figure 1 is used for the fluorescence signal from the sampling cell? In my opinion, this whole paragraph (Line 91-101 along Figure 1) needs more details for clarification.

Line (104): What is so special about polyether ether ketone based valve? Why is it used? What is an advantage for NO?

Line (106): What is the term "servo controlled"? Please describe the functionality/advantage.

Line (108): How much time is required to achieve stability within 1% of set points? This could be a problem during an ascend/descend (changing altitude) of flight measurements. Did the system manage to keep up with the ascending/descending of the aircraft?

Line (109): What is the zero-air? Is it free of NO?

Line (125-132): Why the comparison between 215 nm and 226 nm. It needs motivation.

Line (213-216): Is it a total signal or background-subtracted signal? Do you see any impact of the cell pressure on the zero-air background? Better to present only NO related signal (total signal - background) in Figure 6.

Line (265-272): How the wavelength is tuned (via temperature or current)? Is the laser thermally controlled or what is the impact of an ambient temperature change on the wavelength/power of the laser?

Line (282): Please also show the standard deviation of the average points in Figure 10.

Line (285-294): I think a lot of hydrocarbons/VOCs have absorption in the UV wavelength region (please check at http://satellite.mpic.de/spectral_atlas/). This can lead to a potential interference in particular environments (e.g., forest regions). Can you say something about it?

Line (313): Please provide the confidence interval for the uncertainty.

Line (352): Table 1 shows that the weight of the CL instrument was 150 kg. Generally, commercial CL instruments are very lightweight. What is different in this case?

Line (374): How is the PMT in the CL instrument exposed to cosmic rays at higher altitudes?

Line (377): How many background measurements (frequency) and calibrations are required for the LIF instrument in one day?

Line (382): How much difference in the precision in both cases?

Line (403-404): Maybe I missed the point. Where is the relevant demonstration?

Page 18 (Figure 1): By looking at the layout of the sampling cell, can you say something about dead air pockets near the quartz windows? Are the quartz windows flushed with some dry air? If not, what do you think of dust (build-up on windows with time) related impact on sensitivity or background measurements?

Technical Corrections:

Line (10-15): Please include a reference.

Line (39): "An alternative technique" to "An alternative direct technique".

Line (64): "repitition" to "repetition".

Line (206): Please recheck unit or value "- 90 K".

Line (215): "dependance "to "dependence".

---

## Author Comment (AC1) · 2 Apr 2020

We thank the referee for their useful comments on our manuscript. Below, we address the individual comments. Our responses to the referee's comments are shown in **bold**.

––––––––––––––––––––

L23: while it may be true that very low NO chemistry remains poorly understood, it is known that many of the OH observations used in the Rohrer 2014 study suffered from positive artifacts. So, not sure if this is the best reference. Perhaps there are more recent observations that are artifact-free and illustrate this point (e.g. from SOAS or

GOAMAZON)?

**We have removed the Rohrer citation. We revised this sentence and instead cite the Fittschen (2019) study which provides a comprehensive discussion of this issue.**

L64: "repetition"

**Fixed.**

L178: What is the actual width of the fluorescence collection gate (in ns) used in data acquisition? Does it exclude the laser pulse?

**We have added some text at the end of this paragraph to describe the gate choice (lines 195-199).**

L206: -90K is very cold indeed!

**Fixed, thanks!**

L215: "dependence"

**Fixed.**

L268: what is the width of the running average? And is it just a boxcar window?

**This information was added here (now line 285).**

Sect. 7: Is there any significant background variability in the FIREX data beyond what is seen in the lab? If so, it seems like this would affect the chosen background smoothing window.

**A line addressing this point was added to section 6.1 (lines 359-361). A paragraph in Section 7 is also added to discuss this issue (now lines 374 – 381).**

L373 – 378: These statements would fit better in Section 7.

**These sentences have been moved to the end of Section 7.**

L404: Bringing up the isotopologue detection here seems out-of-place. While this is exciting, it might be better to state that this is possible rather than to state that you have done it (unless you want to show some data to support it).

**Agreed. We have modified this sentence state that it will be the focus of future work.**

Figure 9: Caption incorrect.

**This caption has been corrected.**

---

## Author Comment (AC2) · 2 Apr 2020

We thank the referee for their useful comments on our manuscript. Below, we address the individual comments. Below our responses are shown in **bold**.

———————————————

Specific comments: Line (1-8): Please provide key numbers/values (e.g., wavelength, uncertainty, etc.) in the abstract.

**These values have been added to the abstract.**

Line (37): What is the function of dry ice in a CL instrument? Please can you elaborate

or provide a reference?

**A statement has been added here describing the use of cryogen for CL (now lines 39-40).**

Line (44-45): How do you know it is red-shifted (Measured or from literature)?

**Here we are referring to the specific work of Bradshaw et al., and details of that work are described in the reference which was included.**

Line (63): Why is it important to resolve Doppler broadened in the case of the LIF technique?

**We had stated here that resolving the Doppler broadened spectrum allows us to 'achieve high signal levels and distinguish the NO isotopologues'. To further clarify, we added a sentence in section 2.3 (now lines 174-175) specifically comparing to the effective absorption cross section reported by Bradshaw et al. who used a laser with a wider spectrum (their effective cross section was about 8 times smaller).**

Line (66): Is the detection limit at 10 seconds measured or calculated by simply dividing 1 ppt with sqrt (10)? How do you know it follows the square root dependency?

**This is shown from our measurements in Figure 12, and Figure 11 shows that the precision follows Poisson counting statistics.**

Line (79-82): The sentences "The DFB output. . ...an end capped fiber" are not easily understandable. Describe the purpose, why is this performed or what is required to achieve? Please modify it for simplicity.

**We prefer to not modify this text. Describing in complete detail the fiber laser system is beyond the scope of the current paper. Instead we chose to in this paper to provide a very brief description and a few relevant specific parameters for this system. Many other papers on fiber lasers describe these systems in**

**more detail (including our earlier paper which was cited here, Rollins et al., 2016).**

Line (94): What is the purpose of the reference cell? Why is it necessary to have 500 ppb of NO? This needs a further explanation for a reader.

**The use of the reference cell was described in section 3 (lines 265 – 272). We added a sentence in the section that the reviewer refers to (now lines 106 – 108) to clarify the choice of NO concentration in the reference cell.**

Line (96): What is the absolute pressure? Is there an orifice involves or just pump is used to achieve a lower pressure inside both cells?

**A sentence here has been added to state the absolute pressure (now lines 98-99), and it was also already addressed in the subsequent sections. The paragraph following this one already discussed the system that is used to control the pressure and flow (was lines 102-109).**

Line (97-101): It is not clear from Figure 1 the location of the fused silica lens and the bandpass filter? What type of signal comes from the reference cell (I assume absorption)? Which of the PMT in Figure 1 is used for the fluorescence signal from the sampling cell? In my opinion, this whole paragraph (Line 91-101 along Figure 1) needs more details for clarification.

**We had used 'fused silica' in the text and 'quartz' in the figure. Now it is stated as 'quartz' also in the text.**

**The two dimensional drawing (Figure 1) does not actually show the location of the PMTs. We have changed the main text and the figure caption to describe this (lines 102 – 104).**

Line (104): What is so special about polyether ether ketone based valve? Why is it used? What is an advantage for NO?

**A sentence about the choice of PEEK has been added (lines 111-112).**
Line (106): What is the term "servo controlled"? Please describe the functionality/advantage.

**The word "servo" was eliminated here to avoid confusion.**

Line (108): How much time is required to achieve stability within 1

**We intended for that statement to indicate that the system was able to keep pressure and flow to within that range during all flight maneuvers. This sentence has been modified for clarity.**

Line (109): What is the zero-air? Is it free of NO?

**We have now specified the NO mixing ratio (<2 ppt) in the zero air.**

Line (125-132): Why the comparison between 215 nm and 226 nm. It needs motivation.

**We added the citation here to the Bradshaw et al. work which used 226 nm excitation.**

Line (213-216): Is it a total signal or background-subtracted signal? Do you see any impact of the cell pressure on the zero-air background? Better to present only NO related signal (total signal - background) in Figure 6.

**This is only the photons from NO LIF, not any photons from background. We changed the word "signal" to "sensitivity" here and in Figure 6 to clarify.**

Line (265-272): How the wavelength is tuned (via temperature or current)? Is the laser thermally controlled or what is the impact of an ambient temperature change on the wavelength/power of the laser?

**This had been stated in the instrument description section (line 77). We have added a sentence here to re-state this (lines 276-277).**

Line (282): Please also show the standard deviation of the average points in Figure 10.

**We have added error bars showing the uncertainty in the mixing ratio delivered**

to the instrument for each calibration point. Showing the standard deviation in the signal is not appropriate here, though we could show the standard error. The standard error however is < 0.5% for each data point, and is too small to see on the figure. We have added a statement to the figure caption indicating this.

Line (285-294): I think a lot of hydrocarbons/VOCs have absorption in the UV wavelength region. This can lead to a potential interference in particular environments (e.g., forest regions). Can you say something about it?

**A paragraph has been added to Section 7 discussing the effect of species other than NO on the observed signal (lines 374 – 381).**

Line (313): Please provide the confidence interval for the uncertainty.

**Done.**

Line (352): Table 1 shows that the weight of the CL instrument was 150 kg. Generally, commercial CL instruments are very lightweight. What is different in this case?

**Commercial instruments have much less sensitivity / time response. A sentence has been added to state this (lines 368 – 369).**

Line (374): How is the PMT in the CL instrument exposed to cosmic rays at higher altitudes?

**It is not practical to block the cosmic rays, which pass through many materials including aluminum. A few words have been added here to indicate this (line 398).**

Line (377): How many background measurements (frequency) and calibrations are required for the LIF instrument in one day?

**The LIF instrument does not require a periodic NO free 'zero' measurement as does the CL instrument. Rather, just the difference between online and offline measured at 10 Hz determines the NO mixing ratio. This was described in previ-**

**ous sections, but now we re-emphasize it here (lines 410-413). The frequency of calibrations and variability in those was described in Section 6.**

Line (382): How much difference in the precision in both cases?

**The noise increased by as much as 3x. This is now stated here.**

Line (403-404): Maybe I missed the point. Where is the relevant demonstration?

**We have eliminated this statement.**

Page 18 (Figure 1): By looking at the layout of the sampling cell, can you say something about dead air pockets near the quartz windows? Are the quartz windows flushed with some dry air? If not, what do you think of dust (build-up on windows with time) related impact on sensitivity or background measurements?

**A small amount of flow is exhausted through the arms of the LIF cell to help with these issues. This is now stated in the text (line 115) and also in the figure caption.**

Technical Corrections: Line (10-15): Please include a reference.

**A reference has been added.**

Line (39): "An alternative technique" to "An alternative direct technique".  Line (64): "repitition" to "repetition".

**Done**

Line (206): Please recheck unit or value "- 90 K".

**Thanks, this typo has been fixed.**

Line (215): "dependance "to "dependence".

**Fixed.**